



# Comparison of different cross-sectional approaches for the structural design and optimization of composite wind turbine blades based on beam models

Edgar Werthen[1], Daniel Hardt[1], Claudio Balzani[2], and Christian Hühne[1]

[1]DLR German Aerospace Center, Institute of Lightweight Systems, Lilienthalplatz 7, 38108 Braunschweig, Germany
[2]Leibniz University Hannover, Institute for Wind Energy Systems, Appelstrasse 9A, 30167 Hannover, Germany

**Correspondence:** Edgar Werthen (edgar.werthen@dlr.de)

**Abstract.** During the preliminary multidisciplinary design phase of wind turbine blades the evaluation of many design candidates plays an important role. Computationally efficient methods for the structural analysis are needed to cover the required effects, e.g., correct prediction of stiffness matrix entries including the (bend-twist) coupling terms. The present paper provides an extended overview of available approaches and shows their ability to fulfill the requirements for the composite design of rotor blades. Three cross-sectional theories are selected and implemented to compare the cross-sectional coupling stiffness terms and the stress distribution based on different multi-cell test cross-sections. The cross-sectional results are compared with the 2D finite element code BECAS and discussed in the context of accuracy and computational efficiency. The most promising approach achieved a better resolution of the stress distribution compared to BECAS and an order of magnitude less computation time when the same discretization is used. The deviations of the stress distributions are below 10 percent for the most test cases. The results can serve as a basis for the beam-based design of wind turbine rotor blades.

## 1 Introduction

Beam-based approaches are commonly used in the conceptual and preliminary structural design of wind turbine blades. They are often embedded in a multi-disciplinary optimization (MDO) process (Scott et al., 2019; Serafeim et al., 2022) due to their better computational performance compared to high-fidelity finite element (FE) shell- or solid models. A typical application for an MDO is a bend-twist reducing the aerodynamic loads as investigated by Scott et al. (2020) and Bottasso et al. (2012). The blade flexibility affects the angle of attack along the blade and thereby changes the lift and drag force distribution. For the structural optimization, a common objective function is the reduction of mass or costs Lee and Shin (2022).

For larger blades,according to the theory, the mass and costs increase proportionally to the cube of the blade radius, whereas the annual energy production (AEP) increases proportional to the square of the blade radius Gasch and Twele (2011). Faster growing mass and costs compared to the AEP requires the investigation of new technologies to withstand the increased loads and to limit the blade costs as a significant part of the overall turbine costs.



## 1.1 Beam models within the design process of wind turbine blades

The usage of beam models becomes necessary within the structural optimization due to the evaluation of many design candidates in the preliminary design phase. The number of design candidates results from the investigation of different designs for the structural topology (e.g., number and/or of spars) and concepts for materials used and how they are combined in laminate lay-ups, which in turn have to be linked to a manufacturing concept. Consequently, the basic requirement is a significant reduction of the computation time for model creation and the calculation of internal stresses compared to a high-fidelity FE model. The computation time for the stress calculation scales with the number of iterations of the optimization process. For the shell or solid FE model case, variations of the internal structure of the blade, e.g., the spar position, often requires a 3D CAD (Computer added design) model update and the subsequent translation into a new FE mesh. The higher modelling effort and the longer computation times with 3D models are not acceptable in the preliminary design phase. FE beam models require accurate cross-sectional properties as input. In many design processes (e.g., Scott et al., 2019; Wanke et al., 2021), these properties are determined with the help of cross-sectional FE models to derive the mass and stiffness properties and the stress distribution within the cross-section. These 2D FE approaches have the same problems of an expensive model update with re-meshing when the internal structure or layup changes combined with a more expensive solving compared to analytical approaches.

## 1.2 Requirements on an analytical cross-sectional approach

Based on the technologies described above (e.g., bend-twist coupling) that need to be investigated for the design of rotor blades, requirements for an analytical cross-sectional calculation module can be derived. Composite blades are modelled as beams with closed, different single- or multi-cell cross-section geometries that can vary along the beam axis. The parts of the blade e.g., shell panels and spars, consist of different materials. Moreover, different materials within one part can occur. The structure of the blade is mostly thin-walled, except near the blade root and undergo in-plane and out-of-plane cross-sectional deformations. Beside the classical loading of thin beams like bending and extension, shear forces play an important role since they can be dimensioning for the design of rotor blades. The couplings of the beams degrees of freedom due to the inner geometry or the material layup have to be considered for an accurate representation of the blade. The computational efficiency, i.e., fast output with high accuracy, is of high importance as well to allow the assessment of many design candidates in the preliminary design phase. The historical development of cross-sectional approaches for general beam structures is described by Hodges (2006). Chen et al. (2010) compare several existing tools for cross-sectional calculations (e.g., "PreComp" Bir (2006) or "VABS" Yu (2007)).

## 1.3 Target setting

The present paper provides a comprehensive review of available cross-sectional approaches (section 2.4) based on the requirements for the design of composite wind turbine rotor blades formulated above. Three cross-sectional theories are then selected (section 2.5) and implemented to compare the cross-sectional coupling stiffness terms (e.g., extension-torsion, bending-torsion, section 3.2) and the stress distribution in the cross-section (section 3.3). A simple rectangular and a multi-cell airfoil-based





cross-section serve as test cases. The focus is on the shear stress distribution caused by transversal forces since they are more complex to calculate and, among others, dimensioning for rotor blades as already described above. The cross-sectional results are verified with the 2D-finite element code BECAS (Blasques, 2012) and used also by the industry. A verification of BECAS itself with VABS (also a 2D-FE beam cross-section code, (Yu, 2007)) is given in Blasques (2012). The three selected analytical approaches are evaluated with respect to accuracy of the cross-sectional results and the computation time. The best compromise serve as basis for the cross-sectional calculation module of the beam based design tool *PreDoCS* (Preliminary Design of Composite Structures).

## 2 Beam theories

A beam as a mechanical model is characterised by one geometric dimension being significantly larger than the other two. This allows the calculation of the beam, which is a three-dimensional problem, to be divided into a two-dimensional cross-section calculation and a one-dimensional beam calculation. This procedure is also called dimensional reduction (Hodges, 2006) and is described in the following subsections.

### 2.1 Recovery relations between beam and cross-section

The first step is to set up an approach for the kinematic relations of a cross-section, for which the displacements at each point of a cross-section can be calculated from the beam displacements at this point ("recovery relations"). The formulation of the cross-sectional displacements is the core of the cross-sectional theory. In a second step the constitutive relations of the material (material stiffness) is used to calculate the cross-sectional stress distribution from the strains. The cross-sectional stiffness matrix is derived with the principle of virtual work. The integration of the stress and strain distribution describes the internal loads (cutting forces) of a cross-section. Substituting the stresses and strains by the kinematic and the constitutive equations of the material results in the relation between the displacements and the internal loads of the cross-section which denotes cross-sectional stiffness matrix.

### 2.2 1D-FE-Beam

In case of simple load cases (defined by loads and boundary conditions) for beams with only one cross-section, the displacements along the beam can be calculated analytically. For more complex load cases, the 1D-FE beam model, using the cross-sectional stiffnesses from above, has to be solved which results in the displacements of the entire beam. Now the recovery relations from the previous subsection can be used to obtain the cross-sectional displacements at each point of the beam. From then, the strain and stress distribution are calculated according to the cross-sectional theory as described in the previous subsection Hodges (2006).

### 2.3 Degrees of freedom of a cross-section

In general, the cross-sectional stiffness relation of a beam can be written as follows





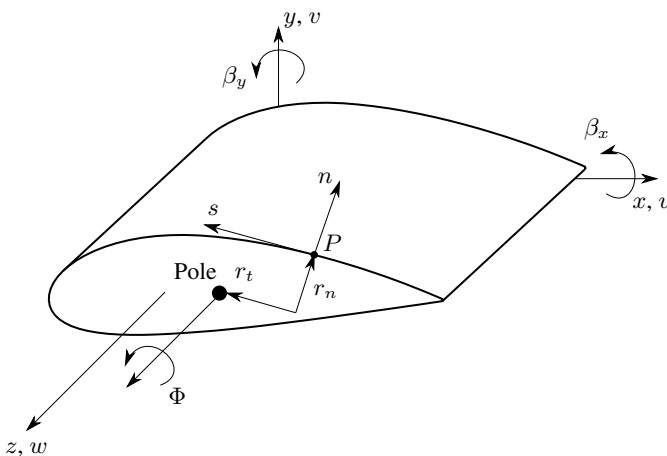

**Figure 1.** Global and contour coordinate system of the beam and cross-section displacements.

$$\boldsymbol{F} = \boldsymbol{K}\boldsymbol{q} \qquad (1)$$

where $\boldsymbol{K}$ is the cross-sectional stiffness matrix, $\boldsymbol{q}$ the vector of cross-sectional displacements and $\boldsymbol{F}$ the vector of the cross-sectional internal loads according to the notation of Jung and Nagaraj (2002). For simplicity, a notation of partial derivatives with subscripts are used, i.e. $\frac{\partial f}{\partial x} = f_{,x}$. Like described by Hodges (2006), depending on the theory, the cross-section has different degrees of freedom (DOF), thus $\boldsymbol{q}$ and $\boldsymbol{F}$ have also a different size and $\boldsymbol{K}$ a different dimension. The DOF are defined with the $z$-axis as beam-axis and the $x$-$y$-plane as cross-section as shown in fig. 1.

In case that only DOF for bending and extension are considered, the Euler-Bernoulli beam model is the simplest and most common. It has three cross-sectional DOF: the longitudinal strain $w_{0,z}$ and the derivative of the twist about the two axes lying in the cross-section plane, $\beta_{x,z}$ and $\beta_{y,z}$. Thus $\boldsymbol{q}$ has a length of 3 and $\boldsymbol{K}$ a size of $3 \times 3$. In order to consider the shear deformation, the Timoshenko beam theory can be used (a first order shear deformation theory, FSDT). The cross-section has then two more DOF: the shear deformation angle in the $x$-$z$-plane $\gamma_{xz}$ and the shear deformation angle in the $y$-$z$-plane $\gamma_{yz}$.

Thus $\boldsymbol{q}$ has a length of 5 and $\boldsymbol{K}$ a size of $5 \times 5$. In addition, the extension and bending part of the theory can be supplemented by a torsional part. The corresponding DOF of the torsional theory are added to the existing ones. St. Venant's theory adds one additional DOF (the twist $\phi_{,z}$), Vlasov's theory would add two additional DOF (the twist $\phi_{,z}$ and the derivative of the twist $\phi_{,zz}$). Different cross-section theories use different numbers of DOF as shown in table 1. For example the approach of Jung (line 5 in table 1) uses a combination of Timoshenko and Vlasovs Theory which results in a stiffness matrix $\boldsymbol{K}$ of size $7 \times 7$,

as further described in section 2.6.





### 2.4 Cross-sectional calculation approaches and their properties

An extensive comparison of cross-sectional theories based on requirements, specifically derived for the design of rotor blades, has been carried out and an extract of this is shown in table 1. Since different assumptions are made for each approach, different effects can be represented. Therefore, based on the described requirements in section 1, the following criteria were selected to compare the approaches in table 1:

- Cross-sectional geometry: open cross-sections, closed single-cell cross-sections, closed multi-cell cross-sections, full cross-sections, thin-walled contour, thick-walled contour

- Calculation approach: analytical approach, 1D-FEM-approach, 2D-FEM-approach

- Considered effects:

    - Cross-sectional stiffness: Number of DOF of a cross-section (dimension of the stiffness matrix), Elastic coupling of the individual DOF of the cross-section, Displacement due to shear force (shear-soft model)

    - Effects due to restrained warping (e.g., occurrence of warping normal stresses)

    - In-plane cross-sectional deformations (e.g., due to transverse contraction)

    - Out-of-plane cross-sectional deformations (e.g., warping due to torsion)

- Material behaviour:

    - Modelling as a disc: constant stresses over the contour thickness

    - Modelling as a plate: non-constant stresses over the contour thickness

    - Consideration of transverse shear (shear in contour thickness direction)

- Force flow in contour direction (circumferential stresses) are equal to zero (common assumptions, not usable for structures under internal pressure, restrained deformation in circumferential direction not considered)

- Linear longitudinal strain distribution over the cross-section, corresponds to the assumptions of the beam models according to Euler-Bernoulli and Timoshenko

The cross-sectional approaches of table 1 can be categorized in different ways. One possibility is the method how the cross-sectional stiffness is calculated.

There are 2D FE approaches, which build up the cross-section from two-dimensional finite element models (Blasques, 2012; Hodges, 2006; Yu, 2007), as well as a mixture of analytical and FE approaches. For the mixed case, an analytical approach is derived to calculate the cross-sectional stiffness, but with the warping of the cross-section assumed as given. This warping is then determined in the second step with a 1D-FE model over the thin-walled contour (cf. Saravia et al., 2015).



**Table 1.** Comparison of existing approaches for the cross-sectional calculation of composite beams

| Reference | Open CS | Closed one-cell CS | Closed multi-cell CS | Thin-walled CS | Thick-walled CS | Full CS | Calculation approach | Dimension of the stiffness matrix | Elastic couplings | Shear-soft model | Out-of-plane deformation | In-plane deformation | Restrained warping | Disc model | Plate model | Transverse shear in constitutive relations | No circumferential stresses | Linear longitudinal strain distribution | Comments |
|---|---|---|---|---|---|---|---|---|---|---|---|---|---|---|---|---|---|---|---|
| Wiedemann (2007) | - | x | x | x | - | - | Analytic | 4 | - | - | x | - | - | x | x | - | x | - | Only for isotropic materials |
| Chandra and Chopra (1992) | x | x | x | x | - | - | Analytic | 9 | x | x | x | - | x | x | x | - | x | - | Extension for multi-cell CS |
| Dugas (2002) | - | x | - | x | - | - | Analytic | 4 | x | - | - | - | - | x | - | - | x | x | Simple approach |
| Johnson et al. (2001) | - | x | - | x | - | - | Analytic | 6 | x | x | - | - | - | x | - | - | x | x | Force-based formulation |
| Jung et al. (2002) | x | x | x | x | x | - | Analytic | 7 | x | x | x | - | x | x | x | x | x | - | Mixed formulation for thick-walled CS |
| Kim and White (1997) | - | x | - | x | x | - | Analytic | 6 | x | x | x | - | - | x | - | x | x | - | Only rectangular CS |
| Libove (1988) | - | x | x | x | - | - | Analytic | - | x | x | - | - | - | x | - | - | x | x | Comparable to Mansfield and Sobey (1979) |
| Mansfield and Sobey (1979) | - | x | - | x | - | - | Analytic | 4 | x | - | - | - | - | x | - | - | x | x | |
| Qin and Librescu (2002) | - | x | - | x | - | - | Analytic | 7 | x | x | x | - | x | x | x | x | x | - | Comparable to Chandra and Chopra (1992), but with transverse shear and 7x7-stiffness matrix |
| Rehfield et al. (1990) | - | x | - | x | - | - | Analytic | 7 | x | x | x | - | - | x | - | - | x | - | Usage of a torsional warping function |
| Weisshaar (1978) | - | x | - | x | - | - | Analytic | 2 | x | - | - | - | - | x | - | - | x | x | Only bending around one axis and torsion |
| Vo and Lee (2008) | - | x | - | x | - | - | Analytic | 8 | x | x | x | - | x | x | - | - | x | - | Extension of Lee (2005) for closed cross-sections |
| Librescu and Song (1991) | - | x | - | x | - | - | Analytic | 7 | x | x | x | - | x | x | x | x | x | - | |
| Song (1990) | x | x | x | x | - | - | Analytic | 7 | x | x | x | - | x | x | x | x | x | - | Comprehensive presentation, taking into account primary and secondary warping |
| Suresh and Nagaraj (1996) | - | x | - | x | - | - | Analytic | 7 | x | x | - | - | x | x | - | - | x | x | Extension of Rehfield et al. (1990) |
| Deo and Yu (2020) | - | x | x | x | x | - | 1D FEM | 4 | x | - | x | - | - | x | x | - | x | - | |
| Saravia et al. (2015); Saravia (2016) | - | x | x | x | - | - | 1D FEM | 6 | x | x | x | - | - | x | - | - | x | - | Warping function with 1D-FE-approach, remaining part is analytical |
| Carrera and Petrolo (2011) | x | x | x | x | x | x | 2D FEM | 6 | x | x | x | x | - | x | x | - | - | - | Only for isotropic materials |
| Giavotto et al. (1983) | x | x | x | x | x | x | 2D FEM | 6 | x | x | x | x | - | x | x | x | - | - | BECAS Blasques (2012), Extension for pre-bended and pre-twisted beams Borri et al. (1992) |
| Hodges (2006) | x | x | x | x | x | x | 2D FEM | 6 | x | x | x | x | - | x | x | x | - | - | Variational Asymptotic Method (VAM) |
| Yu et al. (2005) | x | - | - | x | x | - | 2D FEM | 5 | x | - | x | x | - | x | x | - | - | - | Generalized Vlasov theory (VABS incl. warping) |





The analytical approaches (see table 1 column "calculation approach") can be divided into two categories, the displacement-based formulation and the force-based formulation Jung et al. (2002) which differ in the calculation of the shear stresses. The displacement-based formulation also called stiffness method, has been used e.g. by Rehfield et al. (1990), Song (1990), Chandra and Chopra (1992) and Wiedemann (2007). A displacement field of the cross-section is assumed, from which the shear stresses can then be calculated directly with the help of the constitutive relations. The force-based formulation also assumes cross-sectional displacements and a normal stress distribution is calculated using constitutive relations. Based on the normal stress distribution, the shear stresses are calculated with the help of the integration of the equilibrium condition on a contour element (cf. Jung et al., 2002). The force-based formulation thus leads to better shear stress distributions (Johnson et al., 2001). This approach was used e.g. by Mansfield and Sobey (1979), Libove (1988) and more recently for thin-walled composite beams by Johnson et al. (2001), see table 1. A combination of the displacement- and force based formulation was introduced by Jung and Nagaraj (2002) and is further explained in section 2.6.

## 2.5 Selected approaches based on requirements of wind turbine blades

Based on the requirements described in section 1 and the available approaches listed in table 1, three different approaches were selected. An FE-approach was already excluded due to the high computational cost which will also be shown in section 3.4. From the analytical approaches which fulfill the multi-cell criterion (see table 1) only five approaches remain. The approach of Libove (1988) do not deliver a cross-sectional stiffness matrix. Chandra and Chopra (1992) take into account additional DOF for the derivation of the shear forces which correspond to the line loads. These additional DOF make the approach more complex and are not required for the intended application. The first selected approach for an implementation is the one of Wiedemann (2007) which includes a shear-stiff formulation based on a $3 \times 3$ stiffness matrix (and a torsional stiffness) without bend-twist coupling- and shear stiffness terms. Therefore it does not fulfill all requirements given in section 1 but is chosen anyway due to its fastness and simplicity for implementation. To fulfill the requirements with respect to the elastic coupling and shear stiffness terms, the displacement-based formulation of Song (1990) was chosen as the second approach. It includes furthermore transverse shear and also restrained warping ($7 \times 7$ stiffness matrix). As a third approach the mixed-formulation (displacement- and force based) of Jung and Nagaraj (2002) was chosen due to the expectation that it best fulfills the derived requirements and delivers probably better results in comparison to Song (1990). The three methods are implemented as "Wiedemann", "Song" and "Jung"-approach in *PreDoCS* to create and compare cross-section stiffness matrices and stress distributions. In the following section the theory of the "Jung"-approach is presented.

## 2.6 Theoretical treatment of "Jung's"-approach

The cross-section theory which is used in PreDoCS as "Jung"-approach is derived as described by Jung and Nagaraj (2002). The theoretical treatment is presented in the following, to understand the general approach of determining the cross-sectional stiffness matrix based on the example of the "Jung"-approach. The derived cross-sectional stiffness matrix required for the comparison to other cross-sectional approaches in section 3.

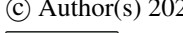



Jung's approach is a so-called mixed approach, or semi-inverse approach, because all element stresses except of the shear stress and the hoop moment can be directly calculated with the given cross-section displacements. The shear stress and the hoop moment are treated as unknowns and are determined by using continuity conditions around each cell of the cross-section.

### 2.6.1 Kinematics

Contrary to Jung and Nagaraj (2002), the $z$-axis is used as the beam axis, not the $x$-axis. This leads to different kinematic equations which are adopted from Librescu (2006). Additionally, the direction of the moment around the $y$-axis ($z$-axis in the Jung coordinate system) is defined the opposite way round. Figure 1 shows the two used coordinate systems. These are an orthogonal Cartesian coordinate system ($x$, $y$, $z$) and a curvilinear coordinate system ($n$, $s$, $z$) at the point $P$, where $s$ is measured along the middle surface of the shell wall and $n$ is normal to $s$. The pole is the pole of rotation around the $z$-axis of

the cross-section and is assumed as given for the derivation of the kinematics.

Assumptions are made for the strains ($\epsilon_{zz}$, $\kappa_{zz}$, $\kappa_{zs}$) of the contour dependent on the cross-section displacements ($w_{p,z}$, $\beta_{x,z}$, $\beta_{y,z}$, $\phi_{,z}$, $\phi_{,zz}$, $\gamma_{xz}$, $\gamma_{yz}$):

$$\epsilon_{zz} = w_{p,z}(z) - x(s) \cdot \beta_{y,z}(z) + y(s) \cdot \beta_{x,z}(z) - \omega(s) \cdot \phi_{,zz}(z),$$

$$\kappa_{zz} = -\beta_{x,z}(z) \cdot x_{,s}(s) - \beta_{y,z}(z) \cdot y_{,s}(s) + r_t(s) \cdot \phi_{,zz}(z),$$

$$\kappa_{zs} = 2 \cdot \phi_{,z}(z). \tag{2}$$

### 2.6.2 Constitutive Relations

To describe the material behavior the CLT (classical laminate theory) is used with the complete coupled 6x6 disc and plate stiffness matrix. Furthermore, the transverse shear stiffness of the plate is also considered. The force and moment flows on an infinitesimal piece of the shell are shown in fig. 2.

The constitutive relations are semi-inverted to obtain the missing force and moment flows and strains ($N_{zz}$, $M_{zz}$, $M_{zs}$, $\gamma_{zs}$, $\kappa_{ss}$) from the strains and flows for which assumptions are given (displacement-based part: $\epsilon_{zz}$, $\kappa_{zz}$, $\kappa_{zs}$; force-based part: $N_{zs}$, $M_{ss}$). It follows for the constitutive relations:

$$\begin{pmatrix} N_{zz} \\ M_{zz} \\ M_{zs} \\ \gamma_{zs} \\ \kappa_{ss} \end{pmatrix} = \boldsymbol{C} \cdot \begin{pmatrix} \epsilon_{zz} \\ \kappa_{zz} \\ \kappa_{zs} \\ N_{zs} \\ M_{ss} \end{pmatrix} \tag{3}$$

### 2.6.3 Determination of $N_{zs}$ and $M_{ss}$

An assumption for $N_{zs}$ and $M_{ss}$ is made in (Jung and Park, 2005, eq. 13, force-based approach):



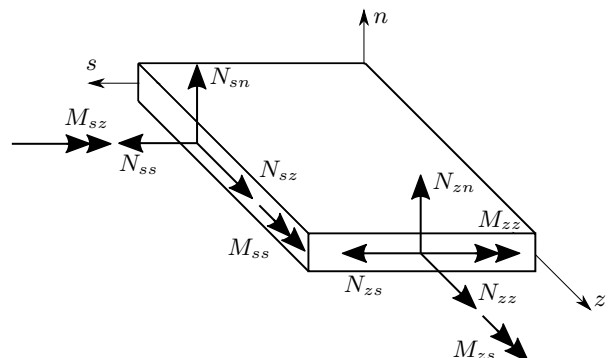

**Figure 2.** Shell forces and moments based on Jung and Nagaraj (2002).

$$N_{zs} = N_{zs}^0 - \int_0^s N_{zz,z}\, ds, \tag{4}$$

$$M_{ss} = M_{ss}^0 + x M_{ss}^x + y M_{ss}^y - \int_0^s M_{zs,z}\, ds, \tag{5}$$

where $N_{zs}^0$, $M_{ss}^0$, $M_{ss}^x$, and $M_{ss}^y$ represent the unknown circuit shear flows and hoop moments for each cell of a closed multi-cell section.

To obtain the continuity condition for each cell of the multi-cell cross-section, four conditions for each cell ($C_i$) are used that are given by

$$\oint_{C_i} \gamma_{zs}\, ds = 2A_i \cdot \phi_{,z}, \tag{6}$$

$$\oint_{C_i} \kappa_{ss}\, ds = 0, \tag{7}$$

$$\oint_{C_i} x \cdot \kappa_{ss}\, ds = 0, \tag{8}$$

$$\oint_{C_i} y \cdot \kappa_{ss}\, ds = 0, \tag{9}$$

where $A_i$ is the enclosed area of the cell $i$.

After solving this linear equation system for the given cross-sectional geometry, the variables $N_{zs}^0$, $M_{ss}^0$ $M_{ss}^x$ and $M_{ss}^y$ are given and the following relationship arises:



$$\begin{pmatrix} N_{zs} \\ M_{ss} \end{pmatrix} = \boldsymbol{\xi} = \boldsymbol{\xi^a} + \boldsymbol{\xi^r} = \boldsymbol{f} \cdot \boldsymbol{q_b} + \boldsymbol{F} \cdot \boldsymbol{q_{b,z}} \tag{10}$$

with $\boldsymbol{q_b} = \begin{pmatrix} w_{p,z} & \beta_{x,z} & \beta_{y,z} & \phi_{,z} & \phi_{,zz} \end{pmatrix}^T$ and $\boldsymbol{\xi^a}$ and $\boldsymbol{\xi^r}$ as the active and reactive parts of the shear flow and hoop moment according to the definition of Gjelsvik and Hodges (1982).

### 2.6.4   Cross-Sectional Stiffness Relations

The cross-sectional stiffness relations are determined in two steps, first the active part and afterwards the reactive part.

**Active part ($\boldsymbol{K_{bb}}$)**

In order to determine the cross-sectional stiffness matrix in the first step, only the active part of the strain energy is considered. With the principle of virtual work, the strain energy related to the virtual strains results in

$$\int_C \left( N_{zz} \cdot \delta\epsilon_{zz} + M_{zz} \cdot \delta\kappa_{zz} + M_{zs} \cdot \delta\kappa_{zs} + N_{zs} \cdot \delta\gamma_{zs} + M_{ss} \cdot \delta\kappa_{ss} \right) ds. \tag{11}$$

Herein, the virtual strains are derived from the virtual cross-section displacements $\delta\boldsymbol{q_b}$. With the help of equation eq. (11), it is possible to establish a relation between the cross-section displacements $\boldsymbol{q_b}$ and the corresponding cross-section loads given

by

$$\boldsymbol{F_b} = \begin{pmatrix} N & M_x & M_y & T & M_\omega \end{pmatrix}^T = \boldsymbol{K_{bb}} \cdot \boldsymbol{q_b}, \tag{12}$$

where $N$ is the normal force, $M_x$ and $M_y$ are the bending moments around the $x$- and $y$-axis, respectively, $T$ is the torsion moment, and $M_\omega$ is the warping bi-moment.

**Reactive part ($\boldsymbol{K_{vv}}$ and $\boldsymbol{K_{bv}}$)**

In order to obtain the shear stiffness terms, a cantilevered beam loaded at the tip with shear forces $V_x$ and $V_y$ is considered. Differentiating $\boldsymbol{F_b}$ with respect to $z$, the following equation can be obtained by

$$\boldsymbol{F_{b,z}} = \begin{pmatrix} 0 & -V_y & V_x & 0 & 0 \end{pmatrix}^T = \boldsymbol{K_{bb}} \cdot \boldsymbol{q_{b,z}}. \tag{13}$$

With $\boldsymbol{q_{b,z}} = \boldsymbol{K_{bb}^{-1}} \cdot \boldsymbol{F_{b,z}}$, the cross-section displacements for extension, bending and torsion, the reactive part of the shear flow $\boldsymbol{\xi^r}$ can be determined using





$$\boldsymbol{\xi^r} = \boldsymbol{F} \cdot \boldsymbol{K}_{bb}^{-1} \cdot \begin{pmatrix} V_x \\ V_y \end{pmatrix} = \boldsymbol{f^r} \cdot \begin{pmatrix} V_x \\ V_y \end{pmatrix}. \tag{14}$$

The shear forces are calculated with the matrix $\boldsymbol{p}$, which is defined by

$$\begin{pmatrix} V_x \\ V_y \end{pmatrix} = \boldsymbol{p} \cdot \boldsymbol{q}. \tag{15}$$

Herein is $\boldsymbol{q} = \begin{pmatrix} w_{p,z} & \beta_{x,z} & \beta_{y,z} & \phi_{,z} & \phi_{,zz} & \gamma_{xz} & \gamma_{yz} \end{pmatrix}^T$ the complete vector of the cross-sectional displacements. The matrix $\boldsymbol{p}$ is splited in a 2x5 left part called $\boldsymbol{p}_1$ and a 2x2 right part called $\boldsymbol{p}_2$.

**Full cross-section stiffness matrix**

Introducing

$$\boldsymbol{K}_{vv} = \boldsymbol{f}^{rT} \cdot \boldsymbol{\Lambda} \cdot \boldsymbol{f}^{rT} \tag{16}$$

and

$$\boldsymbol{K}_{bv} = \boldsymbol{f}^T \cdot \boldsymbol{\Lambda} \cdot \boldsymbol{f}^{rT}, \tag{17}$$

with $\boldsymbol{\Lambda} = \begin{bmatrix} C_{44} & C_{45} \\ C_{45} & C_{55,} \end{bmatrix}$ the resulting 7x7 cross-section stiffness matrix $\boldsymbol{K}$ is obtained. It is given by the expression

$$\boldsymbol{K} = \begin{bmatrix} \left[\boldsymbol{K}_{bb} + 2\boldsymbol{K}_{bv}\boldsymbol{p}_1 + \boldsymbol{p}_1^T \boldsymbol{K}_{vv}\boldsymbol{p}_1\right] & \left[\boldsymbol{K}_{bv}\boldsymbol{p}_2 + \boldsymbol{p}_1^T \boldsymbol{K}_{vv}\boldsymbol{p}_2\right] \\ \left[\boldsymbol{K}_{bv}\boldsymbol{p}_2 + \boldsymbol{p}_1^T \boldsymbol{K}_{vv}\boldsymbol{p}_2\right] & \left[\boldsymbol{p}_2^T \boldsymbol{K}_{vv}\boldsymbol{p}_2\right] \end{bmatrix}. \tag{18}$$

Substitution into eq. (1) yields the relationship between the cross-section displacements $\boldsymbol{q}$ and the internal loads $\boldsymbol{F}$. The order of both vectors is given in table 2.

It should be mentioned that due to the approach for the kinematics, the point of application for extension and bending loads
is the origin of the cross-section coordinate system. The point of application for transversal and torsion loads is the pole (shear center), see fig. 3.

## 3 Comparison of cross-sectional approaches

In this section, the mechanical properties for six different test cases (with two different cross-section geometries) are determined with the help of the three cross-sectional approaches selected in section 2.5. Thereby the stiffness matrix, the positions of the



**Table 2.** Explanation of the indices of the cross-section stiffness matrix $K$, the cross-section displacements $q$ and the internal loads $F$.

| Index | Internal load type |
|-------|--------------------|
| 1 | Transversal force in x-direction |
| 2 | Transversal force in y-direction |
| 3 | Extension |
| 4 | Bending around the x-axis |
| 5 | Bending around the y-axis |
| 6 | Torsion |
| 7 | Warping |

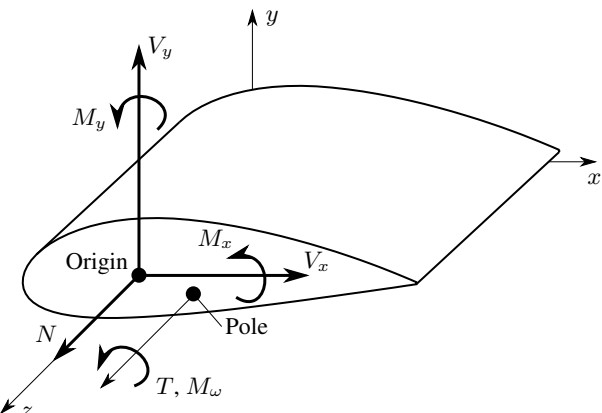

**Figure 3.** Points of attack for the internal loads.

elastic- and the shear center, and the stress distribution across the cross-section are compared. The results of the 2D-FE cross-section solver BECAS (Blasques, 2012) serves as reference.

Both analytical approaches, Song and Jung, require the pole (center of rotation of the cross-section) as input for the kinematic formulations. With the assumption that the center of rotation equals the shear center, the Wiedemann (2007) approach is the only approach that can be used to determine the shear center in advance. In case of Song and Jung the shear center is determined

based on the shear stress distribution.

### 3.1 Test cases

The comparison is carried out using two different cross-sections with different material distributions. One cross-section as a thin-walled rectangle (fig. 4a), allowing a visual verification of expecting stress distributions for simple load cases. The second cross-section is a NACA 2412 airfoil with two shear webs at $30\,\%$ and $50\,\%$ chord shown in fig. 4b, which is representative



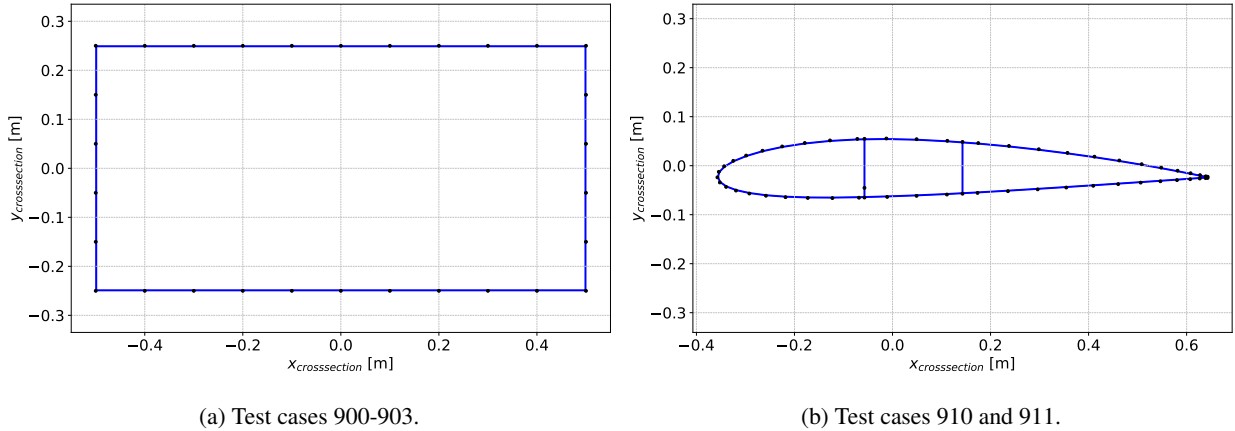

(a) Test cases 900-903.          (b) Test cases 910 and 911.

**Figure 4.** Cross-section geometries (nodes as black circles, elements as blue lines

for a cross-section of a wind turbine rotor blade. For the distinction between effects caused by the geometry and the material, two material concepts are used: Aluminum as isotropic material ($E = 71 \times 10^3\,\mathrm{MPa}$, $\nu = 0.32$) and a composite layup, made up of a carbon fibre UD prepreg based on Hexcel T800/M21 ($E_1 = 134.7 \times 10^3\,\mathrm{MPa}$, $E_2 = 7.7 \times 10^3\,\mathrm{MPa}$, $\nu_{12} = 0.369$, $\nu_{22} = 0.5$, $G_{21} = 4.2 \times 10^3\,\mathrm{MPa}$, $t = 0.184\,\mathrm{mm}$). The stacking sequence of the webs of the NACA 2412 airfoil is $(0/45_2/-45_2/90_2/45_2/-45_2/0)_s$, all other stacking sequences are $(0_2/45/0_2/-45/0_2/45/90/-45/90)_s$. Based on the two cross-sections and the afore mentioned materials the following test cases are created and assigned a unique ID:

- 900: Rect. CS made of $2\,\mathrm{mm}$ aluminum sheets

- 901: Rect. CS made of the composite layup described above

- 902: Rect. CS made of the composite layup described above rotated by $30°$, walls opposite to each other with different sign to get a Circumferentially Asymmetric Stiffness configuration (CAS, Librescu, 2006, p. 91)

- 903: Rect. CS made of the composite layup described above rotated by $30°$ in all walls to get a Circumferentially Uniform Stiffness configuration (CUS, Librescu, 2006, p. 88)

- 910: NACA 2412 CS made of $2\,\mathrm{mm}$ aluminum

- 911: NACA 2412 CS made of the composite layups described above

The contour is discretized in contour direction similarly for all cross-section calculations. The rectangular cross-section (900-903) is discretized in contour direction with 30 equidistant elements, the airfoil with webs (910 and 911) is discretized in contour direction with 58 elements distributed according to the degree of the curvature. The analytical approaches do not need a discretization in contour-thickness-direction, BECAS requires a discretization for each layer of the laminate in contour-thickness-direction. As the laminates consist of 24 layers, 24 elements are used in thickness direction. The resulting number of elements for the different test cases and the different models are listed in table 3.



**Table 3.** Number of elements of the cross-sections for the different approaches.

| Test case ID | Wiedemann, Song, Jung | BECAS |
|:---:|:---:|:---:|
| 900 | 30 | 30 |
| 901 | 30 | $30 \cdot 24 = 720$ |
| 902 | 30 | $30 \cdot 24 = 720$ |
| 903 | 30 | $30 \cdot 24 = 720$ |
| 910 | 58 | 58 |
| 911 | 58 | $58 \cdot 24 = 1392$ |

## 3.2 Stiffness terms

The tables 4 - 7 show the non-zero values of the stiffness matrix and the elastic- (EC) and shear centers (SC) of several test cases. The indices are according to the description given in table 2. The shear stiffness terms of Song show high deviations compared to BECAS. In all test cases deviations above $20\%$ for $K_{11}$ and between approximately $100\%$ and $300\%$ for $K_{22}$ can be observed, due to the first-order shear deformation theory used by this approach. The Jung approach has deviations below $10\%$ which indicates a significant improvement. The Wiedemann approach does not cover the shear stiffness terms due to its shear-stiff formulation. The important coupling stiffness terms show a good accordance with the BECAS results. The stiffness term $K_{36}$ for extension-torsion coupling of test case 902 (CAS) indicated a deviation of about $6\%$. The stiffness terms $K_{46}$ and $K_{56}$ for bend-twist coupling of test case 903 (CUS) indicated a deviation of about $2\%$ to $3\%$. Similar to the shear stiffness, the coupling terms are not present in the Wiedemann approach.

A possible explanation for the slight overestimation of almost all stiffness terms is the different discretization within the tools. The cross-section of PreDoCS is discretized in contour direction only, whereas the BECAS cross-section is additionally

**Table 4.** Test case 901, Rectangular cross-section with composite layup.

| | BECAS | PreDoCS, Jung | | PreDoCS, Song | | PreDoCS, Wiedemann | |
|:---:|:---:|:---:|:---:|:---:|:---:|:---:|:---:|
| | | value | diff. [%] | value | diff. [%] | value | diff. [%] |
| $K_{11}$ | $1.116 \times 10^8$ | $1.155 \times 10^8$ | 3.48 | $1.399 \times 10^8$ | 25.30 | - | - |
| $K_{22}$ | $3.805 \times 10^7$ | $4.225 \times 10^7$ | 11.04 | $9.055 \times 10^7$ | 137.96 | - | - |
| $K_{33}$ | $9.689 \times 10^8$ | $1.045 \times 10^9$ | 7.82 | $1.045 \times 10^9$ | 7.82 | $1.045 \times 10^9$ | 7.82 |
| $K_{44}$ | $4.614 \times 10^7$ | $5.002 \times 10^7$ | 8.40 | $5.002 \times 10^7$ | 8.40 | $5.002 \times 10^7$ | 8.40 |
| $K_{55}$ | $1.281 \times 10^8$ | $1.443 \times 10^8$ | 12.64 | $1.443 \times 10^8$ | 12.64 | $1.443 \times 10^8$ | 12.64 |
| $K_{66}$ | $1.960 \times 10^7$ | $2.102 \times 10^7$ | 7.29 | $2.102 \times 10^7$ | 7.28 | $2.102 \times 10^7$ | 7.28 |
| $K_{77}$ | - | $6.047 \times 10^5$ | - | $6.047 \times 10^5$ | - | - | - |





**Table 5.** Test case 902, Rectangular cross-section with CAS layup.

| | BECAS | PreDoCS, Jung | | PreDoCS, Song | | PreDoCS, Wiedemann | |
|---|---|---|---|---|---|---|---|
| | | value | diff. [%] | value | diff. [%] | value | diff. [%] |
| $K_{11}$ | $1.648 \times 10^8$ | $1.723 \times 10^8$ | 4.52 | $2.024 \times 10^8$ | 22.83 | - | - |
| $K_{22}$ | $5.628 \times 10^7$ | $6.301 \times 10^7$ | 11.95 | $1.227 \times 10^8$ | 118.06 | - | - |
| $K_{33}$ | $6.766 \times 10^8$ | $7.300 \times 10^8$ | 7.89 | $7.300 \times 10^8$ | 7.89 | $7.300 \times 10^8$ | 7.89 |
| $K_{44}$ | $3.227 \times 10^7$ | $3.476 \times 10^7$ | 7.70 | $3.495 \times 10^7$ | 8.30 | $3.495 \times 10^7$ | 8.30 |
| $K_{55}$ | $8.974 \times 10^7$ | $1.008 \times 10^8$ | 12.27 | $1.008 \times 10^8$ | 12.38 | $1.008 \times 10^8$ | 12.38 |
| $K_{66}$ | $2.894 \times 10^7$ | $3.135 \times 10^7$ | 8.33 | $3.135 \times 10^7$ | 8.32 | $3.135 \times 10^7$ | 8.32 |
| $K_{77}$ | - | $3.765 \times 10^5$ | - | $4.225 \times 10^5$ | - | - | - |
| $K_{14}$ | $2.345 \times 10^7$ | $2.494 \times 10^7$ | 6.33 | $2.474 \times 10^7$ | 5.49 | - | - |
| $K_{25}$ | $2.299 \times 10^7$ | $2.619 \times 10^7$ | 13.93 | $2.485 \times 10^7$ | 8.10 | - | - |
| $K_{36}$ | $-4.569 \times 10^7$ | $-4.992 \times 10^7$ | 9.24 | $-4.992 \times 10^7$ | 9.24 | - | - |

**Table 6.** Test case 903, Rectangular cross-section with CUS layup.

| | BECAS | PreDoCS, Jung | | PreDoCS, Song | | PreDoCS, Wiedemann | |
|---|---|---|---|---|---|---|---|
| | | value | diff. [%] | value | diff. [%] | value | diff. [%] |
| $K_{11}$ | $1.708 \times 10^8$ | $1.723 \times 10^8$ | 0.84 | $2.024 \times 10^8$ | 18.51 | - | - |
| $K_{22}$ | $6.147 \times 10^7$ | $6.301 \times 10^7$ | 2.50 | $1.227 \times 10^8$ | 99.66 | - | - |
| $K_{33}$ | $6.983 \times 10^8$ | $7.168 \times 10^8$ | 2.64 | $7.300 \times 10^8$ | 4.53 | $7.300 \times 10^8$ | 4.53 |
| $K_{44}$ | $3.265 \times 10^7$ | $3.331 \times 10^7$ | 2.04 | $3.495 \times 10^7$ | 7.05 | $3.495 \times 10^7$ | 7.05 |
| $K_{55}$ | $8.887 \times 10^7$ | $9.206 \times 10^7$ | 3.59 | $1.008 \times 10^8$ | 13.48 | $1.008 \times 10^8$ | 13.48 |
| $K_{66}$ | $3.064 \times 10^7$ | $3.135 \times 10^7$ | 2.32 | $3.135 \times 10^7$ | 2.32 | $3.135 \times 10^7$ | 2.32 |
| $K_{77}$ | - | $3.874 \times 10^5$ | - | $4.225 \times 10^5$ | - | - | - |
| $K_{13}$ | $-8.932 \times 10^7$ | $-9.143 \times 10^7$ | 2.36 | $-9.983 \times 10^7$ | 11.77 | - | - |
| $K_{23}$ | $-3.107 \times 10^7$ | $-3.344 \times 10^7$ | 7.63 | $-4.992 \times 10^7$ | 60.65 | - | - |
| $K_{45}$ | $9.125 \times 10^5$ | $2.179 \times 10^6$ | 138.76 | 0.000 | $-100.00$ | 0.000 | $-100.00$ |
| $K_{46}$ | $8.005 \times 10^6$ | $8.247 \times 10^6$ | 3.02 | $8.246 \times 10^6$ | 3.01 | - | - |
| $K_{56}$ | $7.868 \times 10^6$ | $8.283 \times 10^6$ | 5.28 | $8.283 \times 10^6$ | 5.27 | - | - |





**Table 7.** Test case 911, NACA 2412 cross-section with composite layup.

| | BECAS | PreDoCS, Jung | | PreDoCS, Song | | PreDoCS, Wiedemann | |
|---|---|---|---|---|---|---|---|
| | | value | diff. [%] | value | diff. [%] | value | diff. [%] |
| $x_{EC}$ [m] | 0.131 | 0.130 | −0.05 | 0.130 | −0.05 | 0.130 | −0.05 |
| $y_{EC}$ [m] | −0.010 | −0.010 | −0.01 | −0.010 | −0.01 | −0.010 | −0.01 |
| $x_{SC}$ [m] | 0.005 | 0.000 | −0.49 | 0.000 | −0.49 | 0.000 | −0.49 |
| $y_{SC}$ [m] | 0.000 | 0.000 | 0.00 | 0.000 | 0.00 | 0.000 | 0.00 |
| $K_{11}$ | $1.061 \times 10^8$ | $1.056 \times 10^8$ | −0.43 | $1.287 \times 10^8$ | 21.40 | - | - |
| $K_{22}$ | $1.509 \times 10^7$ | $1.547 \times 10^7$ | 2.51 | $5.582 \times 10^7$ | 269.80 | - | - |
| $K_{33}$ | $7.454 \times 10^8$ | $7.526 \times 10^8$ | 0.97 | $7.526 \times 10^8$ | 0.97 | $7.526 \times 10^8$ | 0.97 |
| $K_{44}$ | $1.409 \times 10^6$ | $1.408 \times 10^6$ | −0.06 | $1.409 \times 10^6$ | −0.06 | $1.407 \times 10^6$ | −0.17 |
| $K_{55}$ | $7.333 \times 10^7$ | $7.465 \times 10^7$ | 1.80 | $7.465 \times 10^7$ | 1.80 | $7.465 \times 10^7$ | 1.80 |
| $K_{66}$ | $8.043 \times 10^5$ | $8.860 \times 10^5$ | 10.16 | $8.950 \times 10^5$ | 11.29 | $8.828 \times 10^5$ | 9.77 |
| $K_{77}$ | - | $1.553 \times 10^4$ | - | $1.553 \times 10^4$ | - | - | - |
| $K_{12}$ | $-2.308 \times 10^5$ | $-2.923 \times 10^5$ | 26.68 | $-2.878 \times 10^5$ | 24.70 | - | - |
| $K_{34}$ | $-7.445 \times 10^6$ | $-7.562 \times 10^6$ | 1.58 | $-7.562 \times 10^6$ | 1.58 | $-7.562 \times 10^6$ | 1.58 |
| $K_{35}$ | $-9.749 \times 10^7$ | $-9.807 \times 10^7$ | 0.59 | $-9.807 \times 10^7$ | 0.59 | $-9.807 \times 10^7$ | 0.59 |
| $K_{45}$ | $1.231 \times 10^6$ | $1.245 \times 10^6$ | 1.14 | $1.245 \times 10^6$ | 1.14 | $1.245 \times 10^6$ | 1.14 |

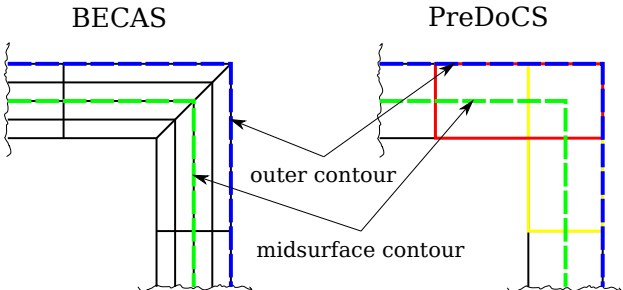

**Figure 5.** Discretization of the contour for BECAS and PreDoCS

discretized in thickness direction. The outer contour of both models is the same. BECAS has a more precised discretization of the contour at the corners. In PreDoCS overlapping elements occur at the corners, which are shown as red and yellow elements on the right-hand side of fig. 5. Due to the overlapping elements the cross-sectional area is overestimated (i.e., excessive material is included in the model) which results in the aforementioned overestimated mass and stiffness terms.



A higher overestimation of about $5\,\%$ to $10\,\%$ can be observed for the torsional stiffness terms $K_{66}$ for all three calculation approaches which indicates a general difference to BECAS. The analytical approaches take an enclosed area for the determination of the torsional stiffness. In general it is recommended to use the mid-surface line of the enclosed contour, shown as green line in fig. 5. Due to manufacturing reasons, PreDoCS uses the outer contour as reference line for the layup definition as

shown in blue in fig. 5. It can be observed, that the enclosed area is overestimated which leads to an overestimated torsional stiffness. The stiffness terms for extension ($K_{33}$) show a deviation to BECAS below $5\,\%$. The bending stiffness terms ($K_{44}$ and $K_{55}$) have a deviation up to $14\,\%$ but only for the rectangular case. This is caused by the overlapping material in the corners. The part of Steiner of the doubled areas leads to non-proportional deviations caused by the square of the distance. The deviations for the elastic and shear center given in table 7 are below $1\,\%$.. It has to be noticed that the stiffness terms for

restrained warping (terms with index 7) are included in the Song and Jung approach but not available in BECAS. Numerical values for warping stiffness terms of closed cross-sections are not provided in literature, neither for beam formulations nor for 2D FE approaches where warping is considered like in VABS (Yu et al., 2005). Therefore, a study on the effect of warping at the level of a beam structure has to be carried out in the future. A comparison of warping displacements and cross-sectional stiffness terms between BECAS and VABS can be found in Blasques (2012).

**3.3 Stress distributions**

Figure 6 and 7 show a qualitative comparison of the shear stress distributions caused by a transverse force in $y$-direction.

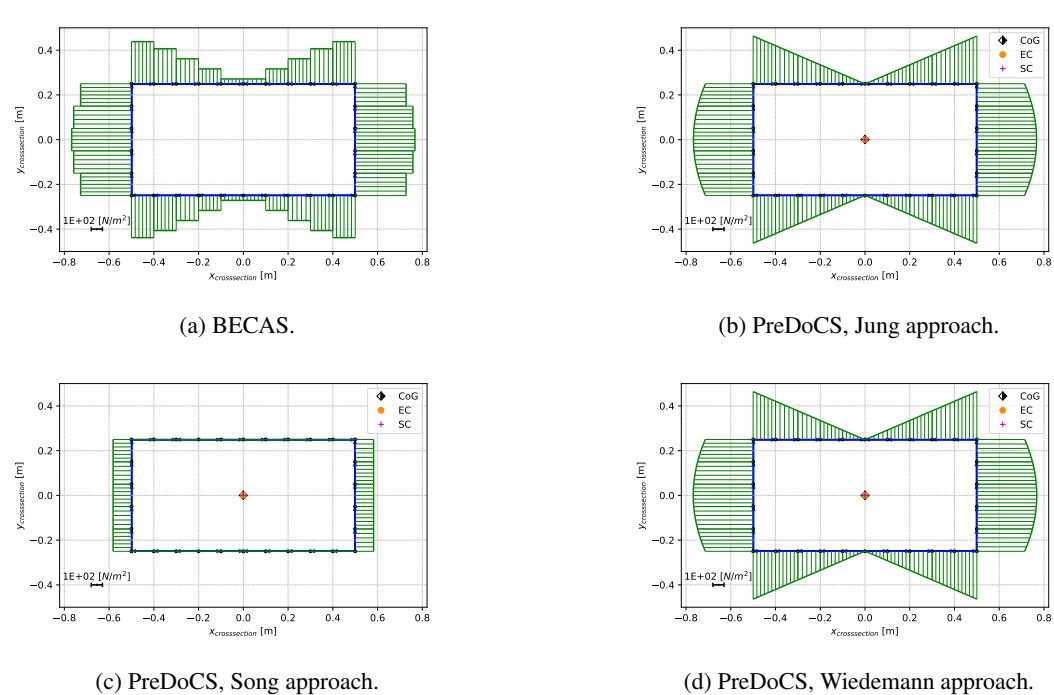

(a) BECAS.

(b) PreDoCS, Jung approach.

(c) PreDoCS, Song approach.

(d) PreDoCS, Wiedemann approach.

**Figure 6.** Shear stress $\sigma_{zs}$ distribution for test case 900 under a unit transverse force in $y$-direction.



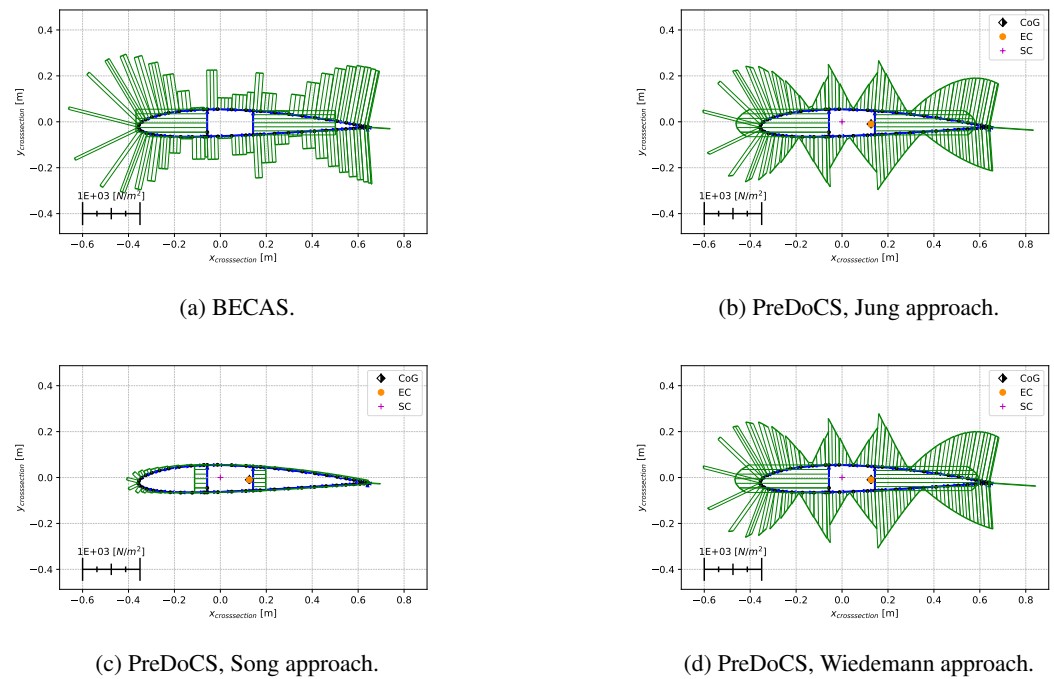

(a) BECAS.

(b) PreDoCS, Jung approach.

(c) PreDoCS, Song approach.

(d) PreDoCS, Wiedemann approach.

**Figure 7.** Shear stress $\sigma_{zs}$ distribution for test case 910 under a unit transverse force in $y$-direction.

A comparison between the three selected approaches for the test cases 900 (rectangular cross-section) and 910 (NACA 2412 airfoil) is shown. Furthermore, the Centers of Gravity (CoG), the Elastic Centers (EC) and the Shear Centers (SC) are displayed. The already mentioned differences in the shear stress distributions can be seen very clearly between the various approaches. The very high deviations of the Song approach result from the usage of the FSDT which assumes a constant shear strain over the entire cross-section. The magnitude of Song's shear stress is only a third of the value of the other approaches. This is caused by Song's assumption that the shear stresses occur in the direction of the contour as well as perpendicular to the contour. Combined with the assumption of the FSDT (constant shear strain over the cross-section) and the isotropic material of the test case 900, the horizontal parts of the contour take 2/3 of the transversal force due to the portion of 2/3 of the total length of the contour. Therefore the vertical parts carry only 1/3 due to the same reason. To illustrate the described effect, the transverse shear stress of the Song approach is shown in fig. 8.

The stress distributions of Jung and Wiedemann show a good accordance with the results from BECAS. A deviation on the webs in test case 910 can be observed (see fig. 7). BECAS shows a constant stress distribution due to the discretization of one element along the web whereas the Jung approach shows a quadratic function due to its exact analytical stress function. It has to be noted that for the same discretization, the resolution of the stress distribution within the cross-section is significantly higher for the analytical approaches, because the stresses (resp. strains) are returned as function of the circumferential contour coordinate, whereas BECAS provides only one stress value per element due to its 2D FE formulation. To provide an exact stress distribution of a rectangular cross-section (as shown in fig. 6), the analytical approaches require only 4 elements (one

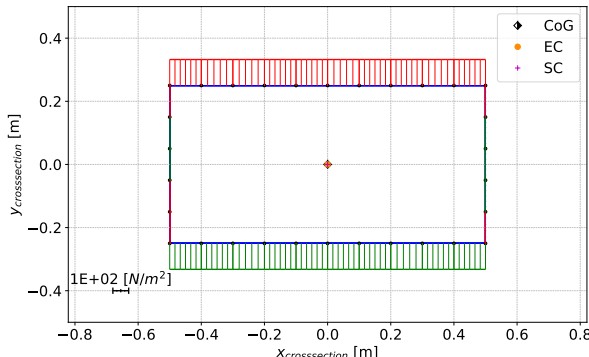

**Figure 8.** Transverse shear stress $\sigma_{zn}$ distribution for test case 900 under a unit transverse force in $y$-direction, calculated with the Song approach.

element per segment line) and can return the stress function or the min. and max. values along one segment. Due to the FE

discretization of BECAS more finite elements are needed to get a correct stress distribution (see fig. 6).

Furthermore a quantitative comparison is carried out and shown in fig. 9. The distribution of $\sigma_{zz}$ and $\sigma_{zs}$ are compared with the distribution of these stresses calculated with BECAS for all test cases and all load cases (transversal force in $x$-direction and $y$-direction, extension, bending around the $x$-axis and the $y$-axis, torsion). Only the active stresses are considered ($\sigma_{zz}$ for extension and bending; $\sigma_{zs}$ for transversal force and torsion), since the reaction stresses become negligibly small and therefore

small absolute differences result in very high relative differences. The goal is to represent the deviation of the stress distribution along the complete path of the cross-section for one load case and one test case in one single value. Therefore the absolute value of the relative difference was formed for each element midpoint of the cross-section, so that negative and positive differences do not cancel out each other. From this "difference distribution" the median is taken as a comparative value, since it is not strongly influenced by local outliers. Outliers can be caused e.g., by discretization mismatches at the corners of the rectangular

cross-section as described in fig. 5. The medians (orange lines) of relative difference for all test case - load case - combinations are shown in fig. 9 as boxplots grouped by test case, load case and cross-section approach.

Figure 9c shows the already mentioned wrongly calculated shear stress distribution under transverse force with the Song approach (due to the FSDT used). The median of the deviation of the Jung theory is below 7.3 %. Outliers can be observed for the test cases 910 and 911 displayed as small circles in the upper right corner of fig. 9a. The corresponding load case

is a transversal load in $y$-direction shown in fig. 9b with the beam at the upper right side. Figure 9c shows that the stress distributions of the Wiedemann theory show a similar deviation for this test cases and load case as those calculated using the Jung approach of about 15 % to 25 % of the median, which needs to be further investigated.





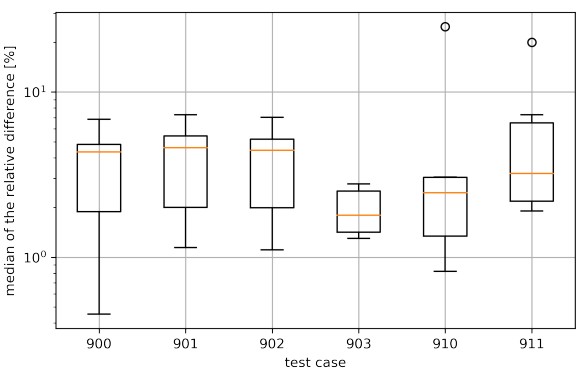

(a) Grouped by test case.

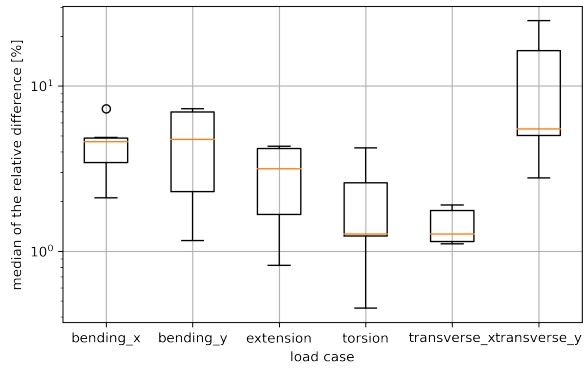

(b) Grouped by load case.

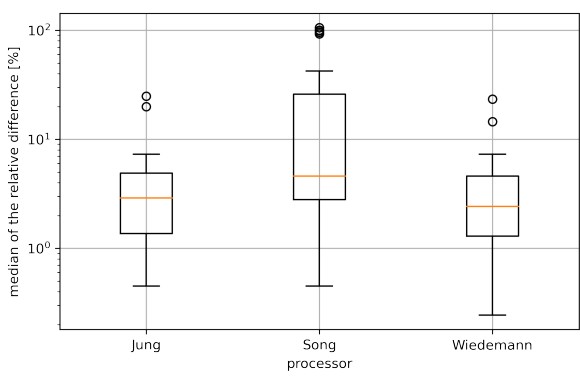

(c) Grouped by cross-section theory.

**Figure 9.** Boxplots of the median of the relative difference for the active normal and shear stress distributions related to BECAS.





**Table 8.** Comparison of the computation time for the calculation of the cross-sectional properties and one load case, compared to BECAS.

| Approach | Cross-section | | | Load Case | | |
|---|---|---|---|---|---|---|
| | Mean [ms] | Std. dev. [ms] | Diff. [%] | Mean [ms] | Std. dev. [ms] | Diff. [%] |
| BECAS | 3628.6 | 445.8 | | 71.22 | 48.47 | |
| Jung | 140.0 | 73.0 | −96.1 | 6.43 | 9.33 | −91.0 |
| Song | 96.3 | 43.0 | −97.3 | 1.46 | 0.64 | −97.9 |
| Wiedemann | 37.9 | 17.3 | −99.0 | 17.96 | 130.94 | −74.8 |

### 3.4 Performance

Table 8 shows the computation time for the calculation of the cross-sectional properties for BECAS and the three implemented
cross-section processors in PreDoCS according to the approaches of Jung, Song and Wiedemann. Furthermore the computational time for one load case is displayed. All computations include the time for meshing of the cross-sections. The calculations are executed with the same PC (Win 10 64-bit, AMD Ryzen 7 5800H (8 x 3.2 - 4.4 GHz), 16 GB RAM).

It can be observed that for the cross-sectional calculation the analytic approaches are an order of magnitude (partly even more) faster than BECAS. Also the computing time for a single load case is around an order of magnitude faster than BECAS.
BECAS uses MATLAB which has highly optimized functions for matrix calculations, where a further improvement of the performance is difficult. For the PreDoCS code this optimization of the computation has not been done up to now and it is expected to improve the performance further due to usage of packages like Cython ((Behnel et al., 2011)). Cython provides the option to compile parts of the Python code as native C-like code which can improve the performance significantly.

Furthermore the time savings need to be analysed in the context of a design optimization problem for a complete rotor blade
modelled as a beam. Thereby the PreDoCS module has to provide the stiffness and and stress distributions for multiple cross-sections along the span as shown in fig. 10 and multiple load cases. The gained performance improvement per cross-section and load case will therefore add up.

### 4 Conclusions

The present paper provides an evaluation of different analytical cross-sectional approaches on the basis of requirements derived
for the preliminary design of wind turbine blades. The approaches of Wiedemann, Song, and Jung were used to calculate cross-sectional stiffness matrices and stress distributions across the cross-section. The results were compared to each other and to the 2D FE-based approach of BECAS.

Since transverse forces play an important role within the design of rotor blades, the Song approach turns out as not suitable due to the wrongly determined shear stress distribution caused by the usage of the FSDT. Also the shear stiffness terms and the
360 related coupling terms are not correctly calculated. The Wiedemann approach does not cover the coupling and shear stiffness





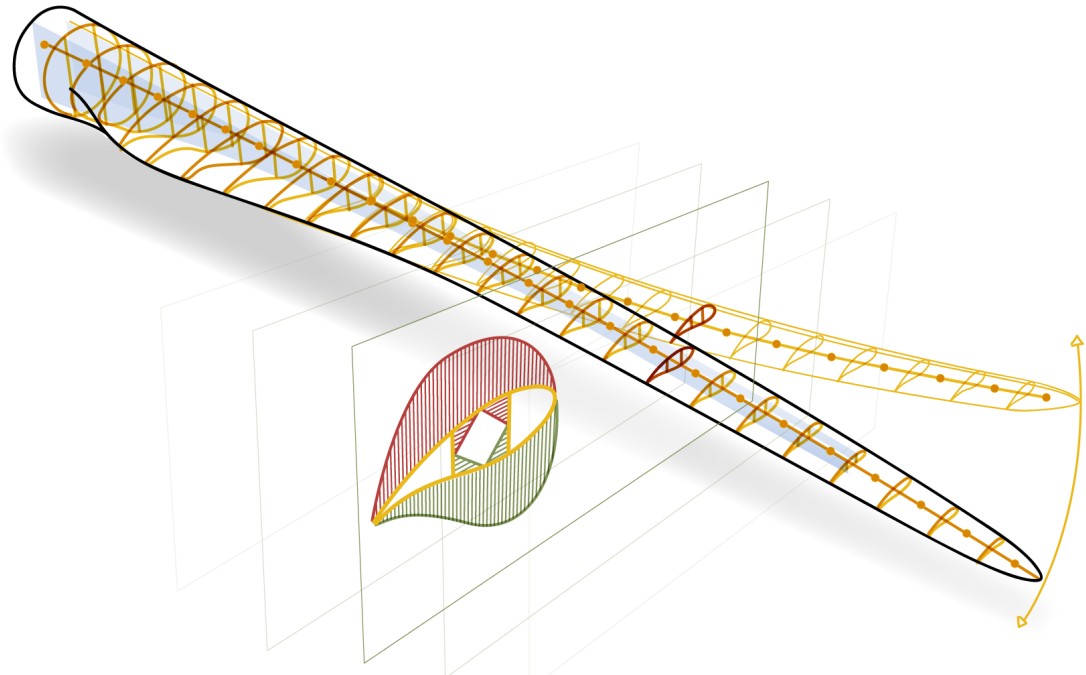

**Figure 10.** Rotor blade with multiple cross-sections.

terms at all. It is a simple and fast approach usable only for determining stress distributions, which show a good accordance with the results from BECAS and Jung.

In terms of accuracy of stiffness terms (also for coupling and shear) and stress distributions, the approach of Jung shows the best results and is therefore taken as cross-section processor in PreDoCS. For test cases 910 and 911 with the transversal load, deviations between $15\,\%$ and $25\,\%$ can be observed in this model. However, it should be noted that the other analytical models do not predict the transverse shear response better.

The comparison of the approaches on the level of a beam structure is work in progress, especially the evaluation of the influence of the shear soft formulation (Jung) on the overall beam deformations. The effect of warping also needs to be further investigated.

The analytical approaches show a significant better performance with respect to computational time compared to the 2D FE code BECAS. This underlines the usability of analytical cross-section approaches in PreDoCS as solver within a design process where many design candidates need to be evaluated. The higher resolution of the stress distribution due to its exact and analytical function of the contour coordinate is easy to differentiate analytically which supports the usage of the approach in gradient-based single- or multi-disciplinary optimization processes with a high number of design variables.



*Code availability.* The usage of the code to create the results of this study is possible upon request to the corresponding author.

*Author contributions.* EW and DH developed the methodology and software, did the analyses and wrote the manuscript. CB und CH revised the manuscript and provided scientific supervision.

*Competing interests.* The authors declare that they have no known competing financial interests or personal relationships that could have appeared to influence the work reported in this paper.

*Acknowledgements.* We would like to acknowledge the funding by the Deutsche Forschungsgemeinschaft (DFG, German Research Foundation) under Germany´s Collaborative Research Center – CRC 1463/1 - Integrated design and operation methodology for offshore megastructures – Project-ID 434502799.



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
