# Peer review of "Comparison of different cross-sectional approaches for the structural design and optimization of composite wind turbine blades based on beam models"

_Wind Energy Science, 2023_

## Referee Comment (RC3)

[referee-annotated manuscript omitted]

---

## Author Comment (AC3)

Dear Julie Teuwen,

We have the pleasure of submitting our revised paper "Comparison of different cross-
sectional approaches for the structural design and optimization of composite wind turbine
blades based on beam models" (wes-2023-147) for consideration in the journal Wind Energy
Science. We are very grateful for the constructive feedback with lots of valuable suggestions
from the editorial team and the reviewers which helped to improve our paper. Based on the
comments from the reviewers, we carried out an extended mesh convergence study and -re-
checked the code for the analytical approaches. We noticed that the enclosed areas were
not determined exactly within the torsional distribution calculation. The changes we made
could significantly improve the results in terms of accuracy. We want to highlight the major
changes and extensions:

• We added an additional beam approach to the overview table 1.
• We clarified the derivation of the shear stiffness terms in Jung's approach (Section
2.6.4)
• We carried out and an extended convergence study for all use cases and approaches
based on a geometrical improved mesh for BECAS and an adapted the cross-section
calculation for the analytical approaches as described above, which significantly
improved the results in terms of accuracy (Section 3.2 and 3.3).
• We added a new figure (figure 8) that shows the stress distribution across the
laminate thickness.
• We updated and extended the performance study with new results (table 8); an
additional line in table 8 shows the performance advantage of the four-element-
model.
• We added a discussion in the conclusion on how geometrical nonlinearity in a blade
beam influences the cross-sectional coupling stiffness terms.
• We gave a clear definition of the terms "elastic center" and "shear center".

Furthermore, we have made all necessary changes and have addressed all comments of the
referees (printed in black) in the detailed response below.

Our response to the referees are written in green.

Reformulated or added phrases for the revised manuscript are referred to in blue.

Line, figure and table numbers in our answers are according to the revised manuscript. Line,
figure and table numbers in the referees' comments are according to the initial manuscript.
New figures and tables are appended to this response. However, the updated figures and
tables were omitted, as this would have been almost all figures and tables. The updated
versions can be seen in the revised manuscript.

We feel that based on the reviewers comments our paper has been sharpened and
improved, especially in terms of clarity, readability, overall language quality, and – in the
authors' opinion – should meet the required standards to be published. If any responses are
unclear, or if you would like to have additional changes implemented, please let us know.

40 Sincerely,

41 Edgar Werthen

42 - On behalf of all authors –

**Referee 1**

Thank you for the constructive and positive feedback. Please find our answers below.

When obtaining the shear stiffness terms, a calculation model is considered with the blade tip loaded. Do you mean that a blade is assumed with the same cross-sections from the root to the tip? Moreover, is the tip load realistic to consider? In fact, a distributed line load is often used when designing the blade.

You are right, a blade certainly consists of several different cross-sections along the blade, and a blade is certainly not loaded by just a single force at the blade tip. The focus of this paper is the calculation of the cross-sectional properties (stiffness and mass matrices) of a beam model, not about a beam model itself. Like in most analytical cross-sectional theories (see e. g. Jung and Nagaraj (2002) equation 23), the approach to integrate the shear stiffness terms in the displacement-based or mixed formulation of the cross-sectional stiffness matrix, respectively, is to assume a prismatic beam with a unit load at the free end, following the first-order shear deformation theory. Once the cross-sectional properties of all cross-sections are calculated, a beam model consisting of several cross-sections certainly needs to be constructed and can subsequently be used to carry out loads simulations, obtaining the real load distribution along the blade. We added the following sentence to section 2.6.4 Cross-Sectional Stiffness Relations:

In order to obtain the shear stiffness terms, a cantilevered beam is considered that is loaded at the tip by shear forces in the x-direction, Vx, and the y-direction, Vy, following the first-order shear deformation theory. It has to be noted that this case does not represent the wind turbine blade use-case. Once the stiffness and mass properties of all cross-sections are calculated, a beam model consisting of several different cross-sections representing the blade certainly needs to be constructed and can subsequently be used to carry out loads simulations, obtaining the real load distribution along the blade.

Secondly, do you assume that the blade will experience relatively small deformations and behave as a linear beam globally? Will the geometrical nonlinearity as a blade beam influence the cross-sectional coupling stiffness terms?

For the calculation of the cross-sectional properties, we indeed apply a linear theory. It is state of the art that the cross-sectional properties are calculated in a pre-processing step applying whatever method, and to assign constant stiffness matrices to the different cross-sections for subsequent turbine simulations. This strategy is not touched in this paper. When the linear cross-sectional properties are used to set up a beam model, the beam model itself must include geometrical non-linearity in the sense of large deflections, as blades undergo very large deflections in operation. Large deflections in turn result in additional coupling effects. For example, when considering equilibrium in the deformed state (which is the definition of geometrical non-linearity), large flap-wise deflections trigger edge-wise bend-twist coupling, which is not accounted for in a linear beam theory. As mentioned above, this paper is about the calculation of cross-sectional properties. However, if geometric nonlinearity would be included in the beam kinematics, the cross-sectional displacements would become non-linear as well. However, this interaction would have to be included in a beam formulation and could not be addressed in a pure pre-processing step. In this case, the cross-sectional properties would need to be updated in each iteration of the beam solution. This is far beyond the scope of this paper, but would definitely be interesting to look at. Maybe this could be done in future work. We added the following sentences as discussion in the outlook section of the paper:

In general, the beam model itself must include geometrical non-linearity in the sense of large deflections, as blades undergo very large deflections in operation. Large deflections in turn result in additional coupling effects. For example, when considering equilibrium in the deformed state (which is the definition of geometrical non-linearity), large flap-wise deflections trigger edge-wise bend- twist coupling. If geometric non-linearity would be involved in the beam theory applied for the
calculation of the cross-sectional properties, the structural parameters of the cross-sections would
need to be updated in each iteration step of the non-linear beam solution, i. e., in each iteration of
each time step in the turbine simulation. This could potentially affect the turbine dynamics, which
would be interesting to look at. However, this goes far beyond the scope of this paper and may be
subject of future work. In any case, such extension would make the turbine simulation very costly, as
the number of iterations would increase dramatically.

Only one blade cross-section is considered. It might be interesting to consider at least two cross-
sections with different aerodynamic profiles.

We have two different profiles. One rectangle, allowing a visual verification of expectable stress
distributions for simple load cases and a NACA 2412 with two shear webs representing a rotor blade.
Adding the material combinations included in the paper, 6 different variants were created in total,
which - in the opinion of the authors - is enough to conclude that the method generally works and
that the comparison is fair and reasonable. This said, we would like to emphasize that we are
principally open to add another application example, but it is not clear what type of cross-section
would really add value and insight instead of just extending the length of the manuscript. We would
therefore prefer to stay with the treated cross-sections and hope for your agreement.

**Referee 2**

We thank the referee for the constructive feedback. Please find our answers below.

The paper is dotted with typos, grammatical and formatting imprecisions. I would strongly
recommend another round of careful proofreading.

Thank you for the hint. We did another round of careful proofreading and hopefully removed all
grammatical and formatting imprecisions.

It would be good to add some comments into the manuscript on whether or not the predictions of
the stiffness coefficients are mesh insensitive or, in other words, to have more details in the paper
about the meshes adopted. It would also be good to have figures showing the meshes, especially
around geometric details. This is because there is evidence in the literature that BECAS and VABS
predictions are mesh sensitive, with fine meshes and accurate geometric representation of the cross-
section being required for accurate results. See, e.g., https://wes.copernicus.org/preprints/wes-
2023-85/. So, are the BECAS reference solutions converged?

We geometrically improved the mesh and performed a mesh convergence study for BECAS and the
analytical approaches for all use cases. Furthermore, we re-checked the code for the analytical
approaches and noticed that the enclosed areas were not determined exactly within the torsional
distribution calculation.  The changes we made, could significantly improve the results. Since the
BECAS results are now presented based on the converged mesh, they can be interpreted as accurate.
For the analytical approaches the same mesh discretization in contour direction is used to be able to
compare the stress distributions.  The stiffness deviations could be reduced from 15% to 5%. Stress
deviations of single outliers could be reduced from 25% to 12%. In most cases, the deviation is far
below the aforementioned numbers. The numbers of elements for both meshes are given in table 3.
We extended figure 4 to show cutouts of critical points for the discretization of the BECAS mesh like
the edge of the rectangular cross-section and the web-shell interaction point of the NACA profile. We
added the following sentences to section 3.1 Test cases and extended figure 4 given in appendix:

To obtain accurate results for BECAS, a fine mesh and an accurate geometric representation of the
cross-section is required (Maes et al., 2024). The contour is discretized in contour direction similarly
for all cross-section calculations based on a mesh-convergence study. The rectangular cross-section
(0-3) is discretized in contour direction with 300 equidistant elements of 10 mm length. It should be mentioned that for the rectangular cross-section the analytical approaches are independent of the
discretization and already obtain accurate results with a discretization of four elements in contour
direction. A further discretization refinement does not affect the calculation results. Nevertheless, in
order to be able to compare element-wise stresses, the same discretization in contour direction was
chosen for the analytical approaches and for BECAS. The airfoil with webs (test cases 10 and 11) is
discretized in contour direction with 225 elements of 10 mm length. The analytical approaches do
not need a discretization in contour-thickness-direction, BECAS requires a discretization for each
layer of the laminate in contour-thickness-direction. As the laminates consist of 24 layers, 24
elements are used in thickness direction. The resulting number of elements for the different test
cases and the different models are listed in table 3.

A similar comment applies to the accuracy of the stresses. The authors do comment on the link
between mesh and stress predictions, but it would be useful if that discussion could be expanded.

As mentioned above, the results of BECAS are now presented based on an improved and converged
mesh, therefore they are treated as accurate. Furthermore, the cross-section calculation for the
analytical approaches could be improved. As already mentioned above, stiffness and stress
deviations could be significantly reduced. The analytical approaches also need an accurate geometric
representation of the cross-section using several linear elements, but the stress distribution is exact
within one element. We extended the discussion on the link between mesh and stress prediction in
section 3.3 "Stress distributions":

The qualitative stress distributions of Jung and Wiedemann show a good agreement with the results
from BECAS. Differences in the absolute values can be observed for test case 10 and will be discussed
later in the qualitative comparison…

…. As already mentioned, for an accurate stress distribution of a rectangular cross-section (as shown
in fig. 5), the analytical approaches require only 4 elements (one element per segment line) and can
return the stress function or the minimum and maximum values along one segment. Due to the FE
discretization of BECAS, more finite elements are needed to get a correct stress distribution (see fig.
5). For cross-sections with segments that are not straight, the analytical approaches also need an
accurate geometric representation of the cross-section using several linear elements, but the stress
distribution is exact within one element.

Similarly, it would be good to see how the different models perform with the stress recovery of all
stress components, not just a few. That's particularly important for the composite models, where
through-the-thickness ply-by-ply stresses are notoriously difficult to resolve.

We created two additional figures to show the stress distribution over the laminate thickness for test
case 1 (rectangular box).  The first figure shows the normal stress, evaluated at the upper part of the
box, under a unit bending moment around the x-axis. The second figure shows the maximum shear
stress, evaluated at the web, under a unit transverse force in y-direction. We added the following
sentences to the manuscript in section 3.3 stress distributions. Please find figure 8 in the appendix.

Fig. 8 shows the comparison of stress distribution along the contour thickness between BECAS (top)
and Jung (bottom) of test case 1 (rectangular cross section with the layup of
$(0_2/45/0_2/-45/0_2/45/90/-45/90)s$). Figure 8a shows the maximum normal stress in longitudinal
direction, $\sigma_{zz}$, under a unit bending around the x-axis, evaluated at the center of the upper edge of
the rectangular cross-section. It can be observed that the 0° plies carry the major portion of the
longitudinal load, which is what the 0° plies are included for. Figure 8 (b) shows the maximum shear
stress $\sigma_{zs}$ under a unit transverse force in y-direction, evaluated at the center of the left web of the
rectangular cross-section. In this case the 45° plies carry the major portion of the shear loads, which
is the purpose of the 45° plies. Both figures show very good agreement between the BECAS and the
Jung solutions.

I would strongly recommend considering an additional beam model that seems to have been omitted
from the paper. Models from the book 'Mechanics of Composite Structures' by Kollár and Springer
have proven useful in various projects, providing accuracy and efficiency.

Thank you for the hint and the reference. We added the very comprehensive approach for to table 1
of the manuscript. However, the approach does not consider the coupling terms, e. g., for extension-
twist or bend-twist coupling. The considered stiffness terms of the approach are listed in the table
below. The bend-twist coupling term is one of the requirements stated in the paper for an analytical
cross-sectional calculation module for wind turbine blades. This requirement is not fulfilled in this
approach. Hence, the model from Kollár and Springer was not included in the calculation
comparisons.

*Table 1: Beam models given in 'Mechanics of Composite Structures' by Kollár and Springer*

**Table 7.2.  Stiffnesses of orthotropic beams with shear deformation**

| Tensile stiffness | $\widehat{EA}$ | Sections 6.2–6.4 |
|---|---|---|
| Bending stiffnesses | $\widehat{EI}_{zz}, \widehat{EI}_{yy}, \widehat{EI}_{yz}$ | Sections 6.2–6.4 |
| Torsional stiffness | $\widehat{GI}_t$ | Sections 6.5.1–6.6.2 |
| Warping stiffness | $\widehat{EI}_\omega$ | Section 6.6.4 |
| Shear stiffnesses | $\widehat{S}_{yy}, \widehat{S}_{zz}, \widehat{S}_{\omega\omega}, \widehat{S}_{yz}, \widehat{S}_{y\omega}, \widehat{S}_{z\omega}$ | Sections 7.2.1–7.2.3 |

I am generally diffident of code-to-code performance comparison. Computer scientists have methods
to do it accurately, but, in an engineering context, so many caveats need to be added that the results
very quickly lose meaning. For instance, for a fair comparison, an accurate baseline needs to be
established. I'd expect the comparison to be done between models that all deliver the same
accuracy, otherwise one may compare, e.g., models that are quick and inaccurate with models that
are slow and accurate. Also, can the authors discern if the speed of each model is related to the
mathematic formulation thereof or to the specific software implementation of that model? More
basically, a computer's OS manages the machine's resources continuously. Comparing run time not
knowing what else the computer was doing during the analysis can be very misleading.

A performance study including several runs to get the standard deviation was performed. Of course
we paid attention, that for all cases, the PC was only focused on the related task and only system
relevant processes were running in parallel. As engineers, these are the options we have. The
expected error should be very small compared to the large difference in the computational time we
see in table 8. We repeated the performance study and updated the computational times and
standard deviations of table 8. The performance study was performed using the same mesh for
BECAS and the analytical approaches. We added an additional line to table 8 (see appendix) showing
the potential of saving computational time when using the analytical approach with a coarser mesh
discretization (that nevertheless shows the same accuracy for the stiffnesses and stress
distributions). We modified the sentences of section 3.4 as follows:

Table 8 shows the computation time for the calculation of the cross-sectional properties for BECAS
and the three implemented cross-section processors in PreDoCS according to the approaches of
Jung, Song and Wiedemann. Furthermore the computational time for one load case is displayed. All
computations include the time for meshing of the cross-sections. For all approaches the same mesh
discretization in contour direction is used (according to table 3) to be able to compare the stress
distributions given in fig. 5 and fig. 6. The calculations are executed on the same PC (Win 11 64-bit,
AMD Ryzen 7 5800H (8 x 3.2 - 4.4 GHz), 16 GB RAM). The analytical approaches achieve a high
accuracy for the rectangular cross-sections already with 4 elements in contour direction. Further
mesh refinement does not affect the stiffness calculation and stress distribution. In contrast to that,
a fine FE mesh is required in BECAS in order to obtain a converged solution. The resulting benefit by
means of computation time savings is shown in the last row of table 8.

**Referee 3**

**General Comments**

The submitted manuscript reviews different methods to determine the cross-sectional stiffness properties of a wind turbine rotor blade on an analytical basis in comparison with BECAS. Since BECAS is a well known tool for this task it perfectly serves as a reference. The advantage of the analytical basis is adequately identified in terms of the calculation speed. This not only serves quick design space investigation but as well high iteration speed in preliminary design. Therefore, the manuscript is of high relevance.

Anyhow, the results show differing deviations from BECAS for bending and torsional stiffness properties depending on complexity of the chosen cross-section. Here the deviation is reduced with increasing complexity. This seems counterintuitive. Thus, more insight into the actual calculation procedure would be helpful. Two of three chosen approaches are only mentioned and not elaborated on, which is why it will be difficult to repeat parts of the work.

All in all, the manuscript is worth of being published with minor revisions. Additional comments can be found in the pdf document attached.

We thank the reviewer for the comprehensive, yet positive and encouraging feedback. In fact, it is correct that the results seemed a bit counterintuitive. Hence, a convergence study has been executed to find a converged BECAS mesh. The mesh could geometrically further improved, especially at the corners.  A quite fine FE mesh was required to obtain the BECAS reference solution that deserves the term "reference". Furthermore, we re-checked the code for the analytical approaches and noticed that the enclosed areas were not determined exactly within the torsional distribution calculation.  With the changes it was found that the solution of BECAS approached the solutions of the analytical approaches. The solutions are much closer now, increasing confidence in the analytical methods. Moreover, the differences are now not dependent on the test case complexity in the sense it was before, meaning the solutions and their deviations are much more intuitive now.

We would like to emphasize that the extended convergence study and the improvements of the cross-sectional calculations of the analytical approaches were triggered by the comments of the referee. We are very thankful for the comments, as the re-calculations improved the quality of the paper a lot and should thus improve its impact on the wind energy research community as well.

**Specific comments**

We integrated smaller changes like wordings, grammatical and formatting imprecisions directly in the manuscript. Your major comments are listed below with our answers:

Which theory? Mass and cost rather scale to the power of 2.3 with radius.

Correct, we changed this and added a reference in the manuscript:

For larger blades, mass and costs increase to the power of around 2.4 with the blade radius (Rosemeier and Krimmer, 2022), whereas the annual energy production (AEP) increases proportional to the square of the blade radius (Gasch and Twele, 2012).

Why Jung only?

Only the approach of Jung is presented, representative also for the other two approaches. The other approaches are documented in literature and are thus available to readers who would like to reproduce our findings. Presenting all three approaches would be far beyond the scope of the paper.
We added the following sentence at the end of section 2.5:

In the following section the theory of the Jung approach is discussed, representative for the other
two approaches as well. The derivation of the other analytical approaches can be found in the
original literature.

May it make sense to set the coordinate system in the order z, s, n to make the composite build up
from outer to inner surface? This would relate to the manufacturing of the blade and easen all
according steps in design.

We took over the coordinate system formulation from other literature, since many publications use
it like that. Treating the cut out shown in figure 2 as cut out from the lower shell the coordinate n
would point inwards. Changing the coordinate system at this stage poses a high risk of errors in the
subsequent parts of the manuscript. Hence, we would like to stick to the coordinate system as it is
and kindly ask for your agreement.

Depending on publication elastic and shear center may be one and the same. Therefore it may make
sense to define both regarding their characteristics.

Correct. Thank you for the comment. We added the following sentences in section 3:

The elastic center is the point where an axial force does not induce bending. The shear center is the
point where applied transverse forces do not induce torsional twist. The presented analytical
approaches use the origin of the cross-section as application point for axial forces and bending
moments. The transverse forces and torsional moments are applied at the shear center.

Manuscript, line 250: "*carbon fibre UD prepreg based on Hexcel T800/M21*"

Referee 3: "This may not be too representative of a wind rotor blade structure"

That is correct. However, we believe that the absolute numbers of the material involved does not
affect the overall outcome and conclusions of the paper, as the study is entirely numerical. Changing
the material will require re-calculation of everything, which would result in a lot of unnecessary
work. Hence, we prefer to stay with this material.

Especially the stiffness terms in longitudinal, bending and torsion show surprisingly high deviations
vs. BECAS. This cannot be explained by missing discretization in thickness direction, only. It rather
hinds towards more general modelling issues like choice of integration path. Since longitudinal
bending is affected as well, this is not just because of the known issue that Ansys depending on the
chosen element has issues to account for the excentricity of the elements.

As already mentioned above, we updated the results based on a geometrical improved mesh and an
extended mesh convergence study for BECAS. Furthermore, we updated the cross-sectional
calculation of the analytical approaches, as described above. The maximum stiffness and stress
deviations could be significantly reduced. The maximum deviation for the stiffness terms of the Jung
approach in longitudinal, bending and torsional direction is now below 1 %. For the used analytical
theory, the integration path should not influence the results. Checks (assert statements) are
integrated in the code along a path to exclude this. We added the following paragraph to section 3.2
Stiffness terms:
The shear stiffness terms of Song show high deviations compared to BECAS. In all test cases
deviations around 20 % for $K_{11}$ and between approximately 100 % and 260 % for $K_{22}$ can be observed,
due to the FSDT used by this approach. The Jung approach shows deviations below 5 % which
indicates a significant improvement for these stiffness terms. The Wiedemann approach does not
cover the shear stiffness terms due to its shear-stiff formulation. The deviations of the main stiffness terms for extension ($K_{33}$), bending ($K_{44}$ and $K_{55}$) and torsion are below 1 % in the Jung approach. The
same applies to the Song and Wiedemann approaches except for test case 3 (CUS layup), where
deviations up to 10 % occur, which have to be further investigated. The coupling stiffness terms show
a good accordance with the BECAS results. The stiffness term $K_{36}$ for extension-torsion coupling of
test case 2 (CUS) is calculated almost exactly. The same applies to the stiffness terms $K_{46}$ and $K_{56}$ for
bend-twist coupling of test case 3 (CAS). Similar to the shear stiffness, the coupling terms are not
present in the Wiedemann approach.

The fact that the deviations comparing to BECAS are reducing when choosing more complex
geometries and layups makes it even worse.
The updated results do not show this trend anymore. We updated the manuscript like mentioned
above.

Manuscript, line 284: "*Due to the overlapping elements the cross-sectional area is overestimated (i.e.,*
*excessive material is included in the model) which results in the aforementioned overestimated mass*
*and stiffness terms.*"

Referee 3: "Why is this not covered?"

The updated results do not show a general overestimation of masses and stiffness terms anymore
due to the more accurate BECAS mesh. We removed the sentences from above. Apart from that, it is
intended to use the presented analytical approach for preliminary design of thin-walled structures.
The overall discretization process is comparable to shell models where similar problems occur. To
avoid this, further adaptions could be included for sharp corners, e. g., at the leading and trailing
edge. These are not implemented yet and are beyond the scope of this paper.

Manuscript, line 291: "*The stiffness terms for extension (K33) show a deviation to BECAS below 5 %.*"

Referee 3: "This is a lot"

The deviations for K33 with the new results is now below 3% and with the preferred approach of
Jung about 1%. Since the intention is to use the Jung approach for preliminary design, deviations
below 1% are acceptable. We added the sentences given above to the manuscript in section 3.2
Stiffness terms and modified the following sentence in the conclusion:

In terms of accuracy of stiffness terms (also for coupling and shear) and stress distributions, the
approach of Jung shows the best results with deviations to BECAS below 5 % (below 1% in most
cases) and is therefore taken as cross-section processor in PreDoCS.

Manuscript, line 291: "*The bending stiffness terms (K44 and K55) have a deviation up to 14 % but only*
*for the rectangular case. This is caused by the overlapping material in the corners. The part of Steiner*
*of the doubled areas leads to non-proportional deviations caused by the square of the distance. The*
*deviations for the elastic and shear center given in table 7 are below 1 %.*"

Referee 3: "The effect occurs as well in non-rectangular cross sections. It is not as prominent but it is
there. Especially when there is high curvature or a corner as in leading edge and trailing edge are,
respectively."

We agree, that the effect is always there, but may be negligible in cases where corners are not sharp.
The updated results, however, after re-calculation subsequently to the aforementioned
improvements, show stiffness deviations for extension, bending and torsion below 1 % for the use-
cases we presented and therefore the mentioned effect of overlapping seems to be negligible. We removed the sentences from above. In general, we agree, that special attention has to be paid for
areas of overlapping like intersections or strong curvatures like in the leading or trailing edge of the
profile.

Manuscript, line 372: "*many design candidates*". Referee 3: "This applies also to design iteration"

Absolutely, we integrated this in the manuscript. Thank you for your comments.

**Appendix:**

[Figure]

(a) Test cases 0-3 (PreDoCS).

(b) Test cases 10 and 11 (PreDoCS).

(c) Test cases 1-3 (BECAS), close-up of the left upper corner.

(d) Test case 11 (BECAS), close-up of the front lower web-shell-intersection.

**Figure 4.** Cross-section geometries (top) and BECAS mesh around geometric details (below).

[Figure]

(a) Normal stress $\sigma_{zz}$ under a unit bending moment around $x$.  (b) Shear stress $\sigma_{zs}$ under a unit transverse force in $y$-direction.

**Figure 8.** Comparison of the stress distribution along the contour thickness between BECAS (top) and Jung (bottom) for test case 1.

**Table 8.** Comparison of the computation time for the calculation of the cross-sectional properties and one load case, compared to BECAS. The second part compares the calculation of the rectangular cross section with a four-element-mesh for PreDoCS.

| Approach | Cross-section | | | Load Case | | |
|---|---|---|---|---|---|---|
| | Mean [ms] | Std. dev. [ms] | Diff. [%] | Mean [ms] | Std. dev. [ms] | Diff. [%] |
| BECAS | 6338.2 | 847.5 | | 428.4 | 282.1 | |
| Jung | 807.1 | 30.3 | −87.3 | 5.06 | 0.95 | −98.8 |
| Song | 592.4 | 37.4 | −90.7 | 4.76 | 0.74 | −98.9 |
| Wiedemann | 321.2 | 13.5 | −94.9 | 3.62 | 0.64 | −99.2 |
| Jung (four-element-mesh) | 8.48 | 0.38 | −99.87 | 0.11 | 0.32 | −99.97 |

---

## Author Response (AR2)

**Referee 2**

I am pleased to see that the authors have taken most of the reviewers' comments constructively onboard. As a result of that, the manuscript has considerably improved. Before I can recommend the paper for acceptance though, I would encourage the authors to take another look at Kollár's work. Specifically, table 6.10 of chapter 6. The model therein is able to produce fully populated stiffness matrices, so I believe the authors are factually wrong stating "the approach does not consider the coupling terms, e. g., for extension-twist or bend-twist coupling."

I fully appreciate that implementing another model from scratch would be a lot of work and I am not demanding that that's done to recommend the paper is accepted, but I would like the authors to have another opportunity to consider Kollár's work. Including that model in the paper (or discarding it with the right justification) would greatly benefit the community.

Dear Reviewer,

we thank you for your comment and we have to apologize for the misinterpretation of Kollár's work. After a careful review of the approach we totally agree that the approach fulfills the requirements for an analytical cross-sectional calculation module (stated in section 1.2) in the same way as the approach of Jung et al. We justify the implementation of Jung's approach with the later possibility of extending it to pre-twisted and pre-bent beams, as given in https://doi.org/10.1163/156855109X428736. This is a necessary step for a later application of the cross-sectional approach in beam models representing a whole wind turbine blade. In section 2.5, we integrated Kollár's work in the decision process where it best fulfills the requirements as that of Jung et al. does. We added the following sentences to section 2.5:

Six analytical approaches fulfilling the multi-cell criterion are available (see table 1)….

Two approaches remain: the mixed formulation (displacement- and force-based) of Jung and Nagaraj (2002) and the force-based formulation of Kollár and Springer (2003). Both approaches are expected to lead to better shear stress distributions in comparison to Song's model (1990). However, Jung's approach was already extended to cover pre-twisted beams (Jung2009), such as wind turbine blades. Since a respective reference for the application of Kollár's model to pre-twisted beams could not be found, Jung's approach was chosen.

We corrected table 1, which now shows that Kollár's work fulfills the same requirements as that of Jung et al.

**Editor**

Dear Julie Teuwen,

Beside the changes we did according to the review of referee 2, we updated the paper for some minor aspects to further improve the quality and the comprehensibility:

37 • We changed the order of the approaches given in table 1 according to the year of
38   publications for the analytical- and for the FE approaches
39 • We added explanations for the order of the stiffness matrix entries given in equation
40   18 (section 2.6.4) to make this point clearer
41 • In section 3.1 Test cases, we changed the reference for the explanation of CUS
42   (Circumferentially Uniform Stiffness) composites to improve the comprehensibility
43 • In Table 6, the stiffness values for K77 were missing in the previous version

44 If any responses are unclear, or if you would like to have additional changes implemented,
45 please let us know.

46 Sincerely,
47 Edgar Werthen

48 - On behalf of all authors –